# Prion diseases disrupt glutamate/glutamine metabolism in skeletal muscle

**Davide Caredio[1], Maruša Koderman[1], Karl J. Frontzek[1,2¤a], Silvia Sorce[1], Mario Nuvolone[1¤b], Juliane Bremer[1¤c], Giovanni Mariutti[1], Petra Schwarz[1], Lidia Madrigal[1], Marija Mitrovic[1], Stefano Sellitto[1], Nathalie Streichenberger[3], Claudia Scheckel[1], Adriano Aguzzi[1]***

**1** Institute of Neuropathology, University Hospital Zurich, University of Zurich, Zurich, Switzerland, **2** Department of Molecular Neuroscience, Weizmann Institute of Science, Rehovot, Israel, **3** Médecin praticien hospitalier en Neuropathologie chez Hospices Civils de Lyon, France

¤a Current Address: Current affiliation: Queen Square Brain Bank, University College of London Queen Square Institute of Neurology, London, England
¤b Current Address: Current affiliation: Amyloidosis Research and Treatment Center, Fondazione IRCCS Policlinico San Matteo and Department of Molecular Medicine, University of Pavia, Italy
¤c Current Address: Current affiliation: Institute of Neuropathology, Uniklinik RWTH Aachen, Aachen, Germany
* adriano.aguzzi@uzh.ch

**Data Availability Statement:** Raw sequencing data as well as processed data from this manuscript is available freely via GEO accession number GSE210128. Code to reproduce results generated

## Abstract

In prion diseases (PrDs), aggregates of misfolded prion protein (PrP$^{Sc}$) accumulate not only in the brain but also in extraneural organs. This raises the question whether prion-specific pathologies arise also extraneurally. Here we sequenced mRNA transcripts in skeletal muscle, spleen and blood of prion-inoculated mice at eight timepoints during disease progression. We detected gene-expression changes in all three organs, with skeletal muscle showing the most consistent alterations. The glutamate-ammonia ligase (*GLUL*) gene exhibited uniform upregulation in skeletal muscles of mice infected with three distinct scrapie prion strains (RML, ME7, and 22L) and in victims of human sporadic Creutzfeldt-Jakob disease. *GLUL* dysregulation was accompanied by changes in glutamate/glutamine metabolism, leading to reduced glutamate levels in skeletal muscle. None of these changes were observed in skeletal muscle of humans with amyotrophic lateral sclerosis, Alzheimer's disease, or dementia with Lewy bodies, suggesting that they are specific to prion diseases. These findings reveal an unexpected metabolic dimension of prion infections and point to a potential role for GLUL dysregulation in the glutamate/glutamine metabolism in prion-affected skeletal muscle.

## Author summary

This study examined how prion diseases, typically affecting the brain, also impact other body tissues. We analyzed gene activity in skeletal muscle, spleen, and blood of prion-infected mice across different disease stages. We found significant gene expression changes, particularly in skeletal muscle. The *GLUL* gene was consistently upregulated in the muscles of prion infected mice and in humans with Creutzfeldt-Jakob disease. This

in this manuscript are available at https://github.
com/marusakod/RML_extraneural_organs. A
searchable database of gene expression profiles
from brain and extraneural organs available for
visualization and download at https://fgcz-shiny.
uzh.ch/priontranscriptomics/.

**Funding:** A.A. is supported by institutional core
funding by the University of Zurich and the
University Hospital of Zurich, a Distinguished
Scientist Award of the NOMIS Foundation, an
Advanced Grant of the European Research Council
(ERC Prion2020 grant ID 278611) and grants from
the GELU Foundation, the Swiss National Science
Foundation (SNSF grant ID 179040 and grant ID
207872, Sinergia grant ID 183563), and the
Human Frontiers Science Program (grant ID
RGP0001/2022). The funders had no role in study
design, data collection and analysis, decision to
publish, or preparation of the manuscript.

**Competing interests:** The authors have declared
that no competing interests exist.

led to disruptions in glutamate and glutamine metabolism, reducing glutamate levels in muscle tissue. These changes were unique to prion diseases and not seen in other neurodegenerative conditions like ALS or Alzheimer's. The findings suggest that prion infections cause specific metabolic disruptions in skeletal muscle, linked to the *GLUL* gene.

## Introduction

Prions are infectious protein aggregates that cause neurodegenerative diseases of the central nervous system (CNS). Prions multiply through the seeded conversion of the physiological cellular prion protein PrP$^C$ into a misfolded, aggregated conformer termed PrP$^{Sc}$ [1]. While PrP$^C$ is primarily expressed in the nervous system, it is also found in skeletal muscle and, to a lesser extent, in lymphoreticular tissue and blood [2,3]. The widespread expression of PrP$^C$ enables PrP$^{Sc}$ propagation at multiple sites in prion infections. Prions can enter the body through the gastrointestinal system and accumulate in lymphoid tissue, leading to neuroinvasion via peripheral nerves [4,5]. Prion seeds are present in the spleen long before the onset of clinical symptoms [6–8]. PrP$^{Sc}$ can also be present in the blood, where it binds to plasminogen [9]. Consequently, blood is a documented route of infection and a significant challenge for transfusion medicine [10, 11]. PrP$^{Sc}$ can be detected in blood with Protein Misfolding Cyclic Amplification (PMCA) as early as 2 months after inoculation [12,13]. However, PMCA is not as sensitive for detecting sporadic CJD (sCJD) [14, 15], which aligns with findings that prion transmission through blood transfusion was reported for vCJD but not for sCJD [16].

Muscle tissue has been a focal point in PrDs due to the potential for significant dietary exposure to prions through meat consumption [17,18]. This concern arises from the potential for prions to be present in muscle tissue, thereby posing a risk of transmission through the food supply. Skeletal muscles of patients with acquired and sporadic CJD show PrP$^{Sc}$ deposits in peripheral nerve fibers [19]. Unlike other organs, prions in skeletal muscle are found only late in disease [6,20]. These findings suggest that the pathogenic processes are systemic and not confined to the brain, providing possible sources for early detection.

Using PrP$^{Sc}$ as a biomarker for early diagnosis of PrDs faces several challenges. For one, the levels of PrP$^{Sc}$ at early disease stages are often too low for detection. Furthermore, distinct prion strains can have unique pathobiological characteristics that influence their presence in peripheral tissues and body fluids [21]. Growing evidence indicates that gene-expression changes in extraneural tissues, including blood, spleen, and skeletal muscle, can serve as markers of neurodegenerative disease progression [22–25]. RNA sequencing of whole blood in Parkinson's disease uncovered early immune cell changes and distinct gene expression patterns [26]. A recent study linked exaggerated type I myofiber grouping in Parkinson's Disease (PD) to altered gene expression in muscle, suggesting significant neuromuscular junction involvement and remodeling [27]. Analogously, differential expression of muscle-specific genes was found in amyotrophic lateral sclerosis (ALS) patients, suggesting muscle-level changes alongside neural degeneration [28]. Lymphoid tissue also accumulates prions [29,30] and may experience molecular changes with diagnostic and prognostic potential.

Here we have conducted transcriptome-wide RNA sequencing analyses on blood, skeletal muscle, and spleen of mice after intracerebral exposure to prions and in autoptic skeletal muscle of humans diagnosed with sCJD. We found that glutamate-ammonia ligase (*GLUL*) is uniquely upregulated in skeletal muscle of prion-infected mice and humans, but not in amyotrophic lateral sclerosis (ALS), Alzheimer's disease (AD), or dementia with Lewy bodies (DLB). This finding, in conjunction with observed reductions in glutamate levels in both

animal models and human cases of prion infection, suggests a disruption in glutamate/glutamine metabolism in skeletal muscles as the disease progresses. This points to a prion-specific muscular pathophysiology diverging from other neurodegenerative disorders.

## Results

### Transcriptional derangement in skeletal muscle during prion disease progression

For this study, we used a previously established cohort of wild-type 2-month old C57BL/6 mice [31] which we had injected intracerebrally (i.c.) with scrapie prions (6th consecutive mouse-to-mouse passage of mouse-adapted Rocky Mountain Laboratory sheep scrapie prions, abbreviated as RML6). For control, we injected non-infectious brain homogenate (NBH). Spleen, hindlimb skeletal muscle and blood were collected during necropsy at 4, 8, 12, 14, 16, 18 and 20 weeks-post-inoculation (wpi) as well as at the terminal stage of disease (Fig 1A). We stratified our collective into three categories: early stage (4 and 8 wpi), pre-symptomatic stage (12, 14, 16 wpi) and symptomatic stage (18, 20, wpi and terminal) [32] (Fig 1A).

We defined differentially expressed genes (DEGs) as transcripts with absolute $\log_2$ fold change $|\log_2 FC| > 0.5$ and p-value $< 0.05$ (Fig 1B). Transcriptional changes in blood were inhomogeneous during disease progression (Fig 1B), possibly because peripheral blood may undergo changes in its cellular composition during infection or inflammation. Nevertheless, in both early timepoints DEGs associated with blood coagulation and hemostasis were detected (S1A and S1B Fig and S1 and S2 Tables). During the presymptomatic and symptomatic stages we did not identify any overlapping changes in blood. In contrast, major transcriptional changes in the spleen were found in terminal disease (Fig 1B). Except for *Pcdh18*, which was significantly altered in both early stage timepoints, there was no overlap between DEGs in blood and spleen (S3 Table).

Compared to other analyzed organs, the number of DEGs in skeletal muscle remained relatively constant throughout the course of the disease (Fig 1B). However, this consistent pattern was punctuated by recurrent up- or downregulation of specific genes at distinct disease stages. During the presymptomatic stage a single gene, *Adh1*, displayed upregulation, whereas the symptomatic stage featured elevated expression of *Mir8114*, *Glul*, and *Pik3r1* (S3 Table). To allow for interactive exploration of the results described in this study and for integration with our previously reported findings [31], we constructed a searchable database of gene expression profiles from brain and extraneural organs available for visualization and download at https://fgcz-shiny.uzh.ch/priontranscriptomics/.

### Lack of post-transcriptional changes in extra-neural organs of prion-inoculated mice

We next calculated the genome-wide adenosine-to-inosine editing index (AEI) to measure global RNA editing levels [33], the preferential site of RNA editing in mammals. Global editing levels in blood rose steadily during aging but were independent of prion inoculation (S2A Fig). No AEI differences were seen in muscle or spleen (S2B and S2C Fig). To determine recoding of individual transcripts, we aligned our sequencing results to previously published high-confidence AEI recoding sites [34]. We found *Flnb* and *Copa* in the spleen and *Cog3* in blood to be significantly recoded (S2D Fig).

Alternative splicing can give rise to disease-associated differentially used transcripts [35]. In contrast to our previous results in the brain [31], the present alternative splicing analyses in extraneural organs showed only minor alterations (S2E Fig). *Necap2*, *Myl6* and *Srsf5*

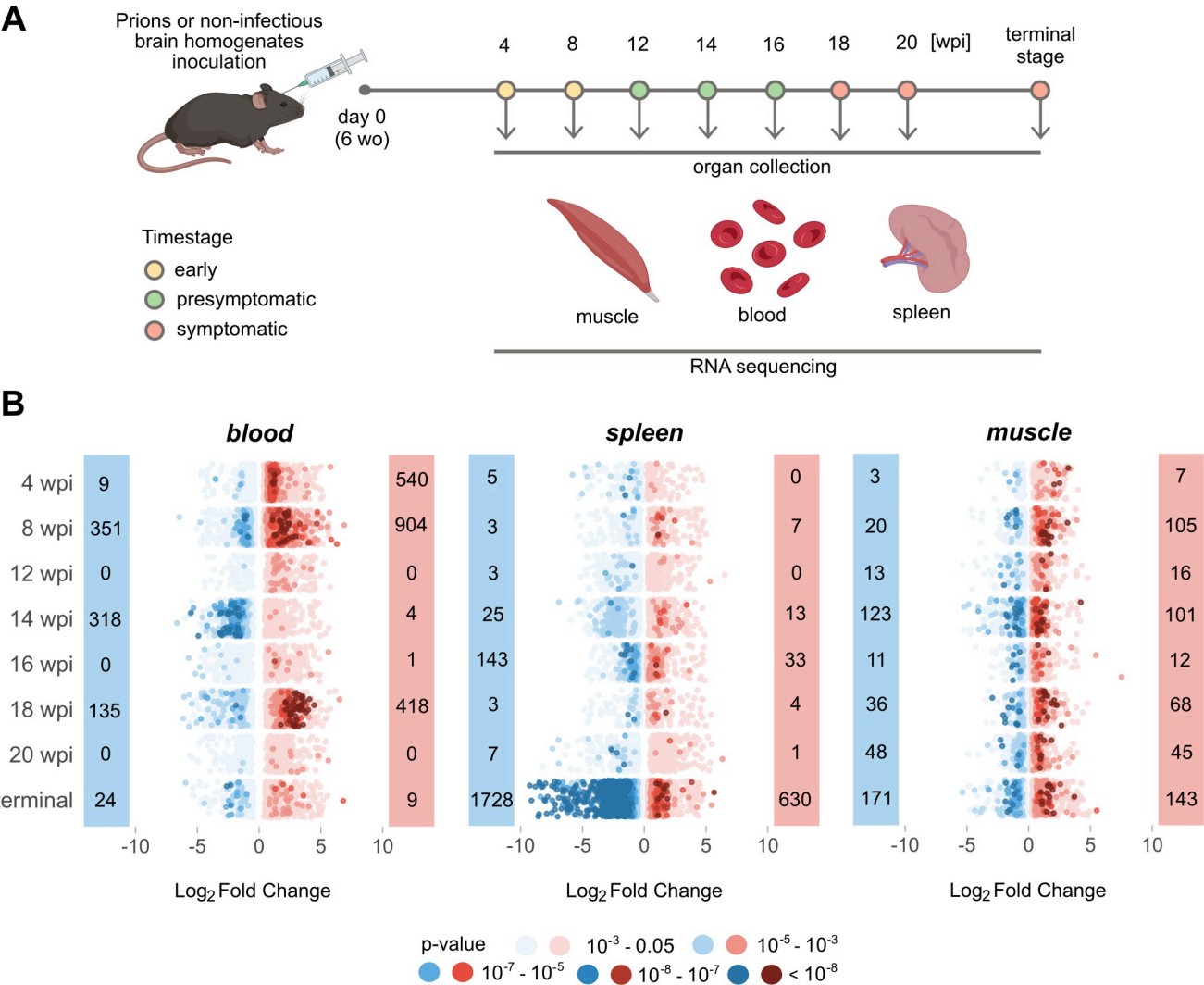

**Fig 1. Temporal Dynamics of Gene Expression in Prion-Affected Tissues. (A)** Muscle, blood, and spleen tissues were collected for bulk RNA sequencing at eight individual timepoints (wo = weeks old; wpi = week post inoculation). Samples were stratified into early, presymptomatic, and symptomatic stages. Panel created with BioRender.com **(B)** Prevalence of upregulated (red) and downregulated (blue) DEGs (p-value < 0.05) across disease progression in the three tissues analysed. The dots in the dot plot represent individual genes and are color-coded according to their corresponding p-values.

transcripts were alternatively spliced across multiple organs and prion incubation times (S4 Table). Only in two out of a total of 21 splice variants differential transcript usage was accompanied by differential gene expression: upregulation of *Myl6* in blood at 4 wpi and downregulation of *Ms4a6c* in blood at 14 wpi.

## Prion Deposits Are Found in Spleen and Skeletal Muscles at All Time Points, with Early Absence in Skeletal Muscles of Mice Infected with RML and Presence in ME7 and 22L Strains

To determine whether these molecular changes are directly caused by on-site prion deposits, we conducted detection assays at various time points using Proteinase K-treated Western Blot (PK-WB) and RT-QuIC assays. We used organs from the same RML prion-infected mice and

corresponding NBH control mice previously used for bulk RNA sequencing, selecting a single time point from each stage: 8 wpi for the early stage, 16 wpi for the presymptomatic stage, and the terminal stage for the symptomatic stage. Blood was excluded from this analysis due to hemolysis during RNA sequencing, which made it impossible to separate and collect plasma for RT-QuIC. Whole blood was unsuitable for analysis because blood cells and their products inhibit the RT-QuIC seeding response [36,37], increasing the likelihood of false negatives. Additionally, blood-derived prions cannot be detected via PK-WB due to low titers.

As shown by other groups [6,38,39], the spleen accumulated prions early in intracerebrally inoculated mice, despite low PrP$^C$ expression, and the infectious agent load remained constant throughout disease progression (S3A and S3B Fig).

The concentration of prions in skeletal muscles is reported to be 10,000 times lower than in the brain, making detection challenging [40,41]. To address this limitation, we used the sodium phosphotungstate anion (NaPTA) reagent with magnesium chloride (MgCl$_2$) to precipitate prions [42]. Consistent with other studies [6,20], we detected prions at 16 wpi and at the terminal stage. However, prion deposits were not observed at the early time point (8 wpi) in mice infected with the RML prion strain. In contrast, prions were detected in the skeletal muscles of mice infected with other prion strains (ME7 and 22L) at all analyzed time points (S4A and S4B Fig). At 8 wpi, prions were detected in mice infected with the ME7 strain in two out of three biological replicates, whereas all replicates showed prion presence in mice infected with the 22L strain. Notably, the PK-WB assay lacked the sensitivity to detect prions at 8 wpi (S4B Fig), whereas RT-QuIC demonstrated significantly higher sensitivity (S4A Fig).

## Consistently altered gene modules in skeletal muscle during prion disease progression

WGCNA (Weighted Gene Coexpression Network Analysis) identifies modules of highly correlated genes, which helps detecting coordinated changes in gene expression. We utilized WGCNA in conjunction with differential expression (DE) analysis to deduce organ-specific gene co-expression networks (S5 Table). To summarize the gene-expression levels of individual network modules, we calculated module eigengenes (MEs) representing the first principal component of each module. We identified 25 and 13 modules in blood and spleen, respectively, but we did not find any significant differences between the MEs of these modules in the two study groups across all three disease time stages (S5A and S5B Fig). Conversely, in the muscle co-expression network, two of 39 modules ("orange" and "darkgreen") showed significant differences in MEs between NBH controls and prions throughout disease progression (S5C Fig). The "orange" module (163 genes) was upregulated, while the "darkgreen" module (198 genes) was downregulated as the disease advanced (Fig 2A).

To better understand these pathophysiological events, we identified genes exhibiting the most notable and consistent expression changes throughout the disease progression within each module of interest. Hub genes were defined by module membership (MM) which is a measure of the correlation between the expression pattern of a given gene and the overall expression pattern of all the genes within the module. Additionally, we derived a gene significance score from p-values obtained using DESeq2 [43]—a tool for identifying expression changes in RNA-seq data—across time stages. Strikingly, the gene significance score was found to be highly correlated (orange module R = 0.81; and darkgreen module R = 0.7) with module membership (Fig 2B). The convergence of two different methods on the same set of genes provides evidence for the robustness of hub gene detection. The top 20 hub genes for orange and darkgreen module are labeled in Fig 2C.

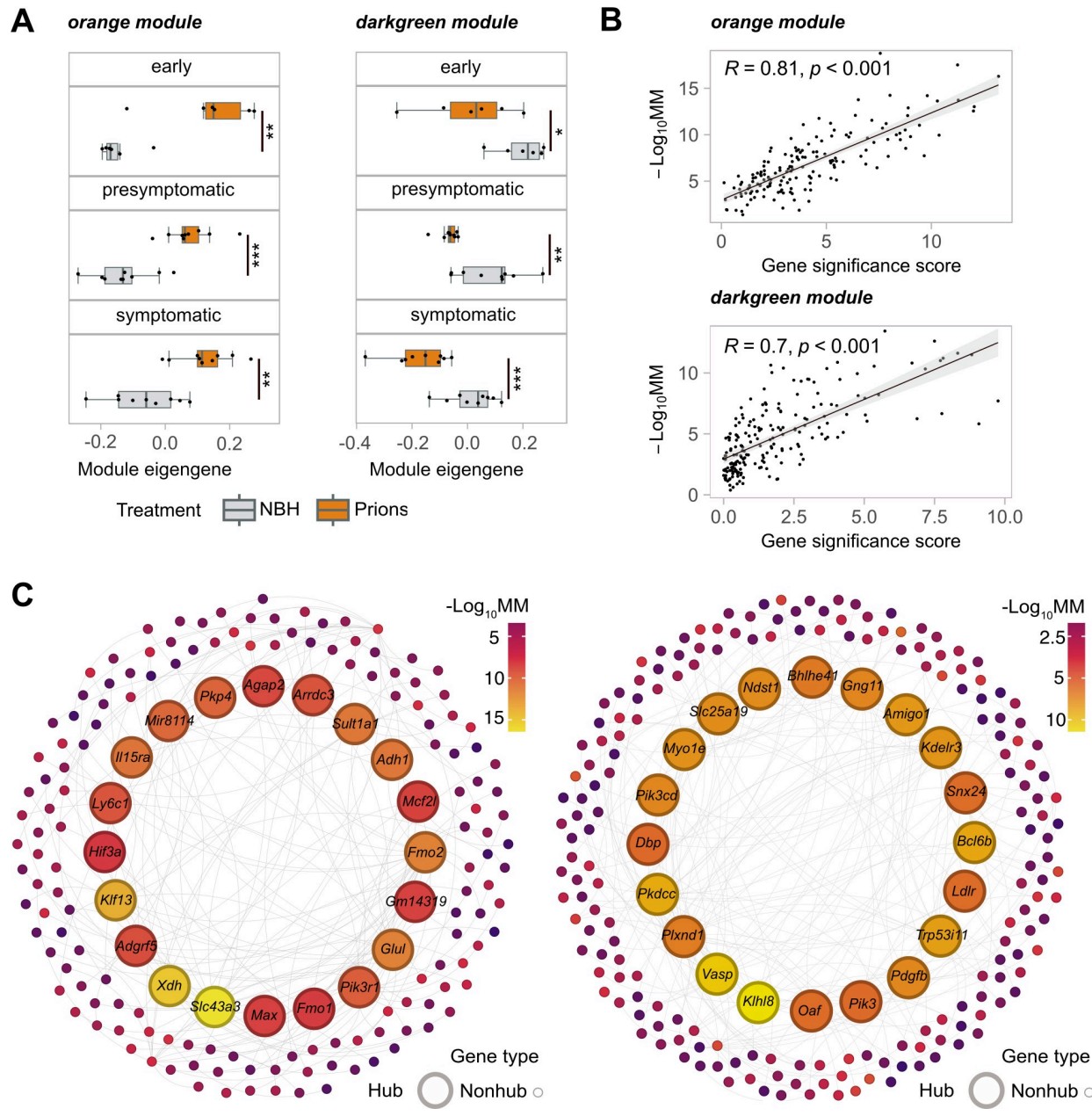

**Fig 2. WGCNA Analysis of Gene Co-expression Modules. (A)** Boxplots of module eigengenes of the main cohort for gene co-expression orange and darkgreen modules identified by WGCNA at different timestages (early, presymptomatic and symptomatic). Statistical significance (*p < 0.05, **p < 0.01, ***p < 0.005, ****p < 0.001) is indicated by asterisks **(B)** The scatter plots illustrate the relationship between the gene significance score and module membership (MM). Pearson correlation coefficient (R) and its corresponding p-value are displayed. **(C)** The minimum spanning trees with nodes representing genes within the orange and darkgreen modules are shown. The colour of each node corresponds to module membership (MM). For each module, 20 hub genes are represented by larger-sized nodes.

## Validation cohort confirms robustness of hub genes for muscle co-expression network in prion-infected mice

To test the reliability and validity of our findings, we investigated a validation cohort comprising samples from each time stage. We first asked whether the muscle co-expression network modules in the main cohort were preserved in the validation cohort. All modules (except for

the "plum", "orangered3", and "brown" modules) showed high preservation with a z-summary statistic >1.96 (S6A Fig). We then calculated the ME for each sample in the validation cohort using the same gene module assignment as in the main cohort. Again, the "orange" and "dark-green" modules were the most affected, exhibiting significant ME differences between RML6 and NBH at both presymptomatic and symptomatic stages (6B Fig). Notably, the trend of ME changes was already evident in the early stage and consistent with the trend observed in the main cohort (Fig 3A).

To further validate our results, we identified hub genes for the validation cohort using MM values that correlated with those of the main cohort (orange module R = 0.55; darkgreen module R = 0.5). The overlapping hub genes between the two cohorts are shown in Fig 4B and 4C. The high correlation indicates that the hub genes identified in the main cohort were robust and reliable in the validation cohort as well. These findings reinforce the idea that the identified hub genes are biologically significant, offering insights into the underlying pathophysiological mechanisms of the disease.

## Upregulation of glutamate-ammonia ligase in skeletal muscles of human and murine prion-diseases

To test whether our findings apply to human PrDs, we performed RNA sequencing on skeletal muscles of sCJD patients (n = 6). For control, we used skeletal muscles of subjects without clinical or pathological diagnosis of neurodegeneration matched for age, gender, and specimen age (n = 4) (see S6 Table for clinical details). The total RNA extracted from the Psoas major muscle exhibited significant degradation (S7A Fig). However, principal component analysis highlighted grouping of sCJD and control tissues in two distinct clusters (S7B and S7C Fig). Based on $|log_2FC| > 0.5$ and FDR < 0.05 (see Methods for details), we identified a total of 365 DEGs, of which 258 were protein-coding (S7 Table).

We then compared the DEGs from sCJD samples with hub genes from the two mouse cohorts, and found only one overlapping gene, glutamate ammonia-ligase (*GLUL*), which was significantly upregulated in human and mouse samples (Fig 3D–3E). We further validated the upregulation of GLUL at the protein level using Western Blot (Fig 3F–3G). We also re-evaluated the upregulation of *Glul* across all individual timepoints in skeletal muscles of prion-infected mice, noting that it remains consistently upregulated. The only exception was at time-point 12 wpi, where the upregulation does not exceed $Log_2FC > 0.5$ (S5D Fig).

While examining post-mortem muscle samples from human patients with sporadic sCJD, we found numerous changes in gene expression that differed from those observed in mouse models. These differences might have only become apparent during the late stages of the disease, where nonspecific changes, possibly influenced by factors like prolonged immobilization, are more likely to occur. However, our analysis of mouse models provides strong evidence that GLUL correlates with the disease process at earlier stages.

## GLUL is upregulated in mice infected with a variety of prion strains

Prion strains are infectious isolates exhibiting distinct biological properties, such as tissue tropism, incubation time, and neuropathological features [44]. Different prion strains may elicit different transcriptional responses in their hosts. To determine whether *Glul* alterations are a common feature across prion strains, we tested whether *Glul* is changed in mice infected with a panel of mouse-adapted prion strains. To this end, we intracerebrally injected an additional cohort of C57BL/6 mice with ME7, 22L, RML6, and NBH, and collected hindlimb skeletal muscle at different time points (8 wpi, 16 wpi and terminal) corresponding to each of the three aforementioned disease stages.

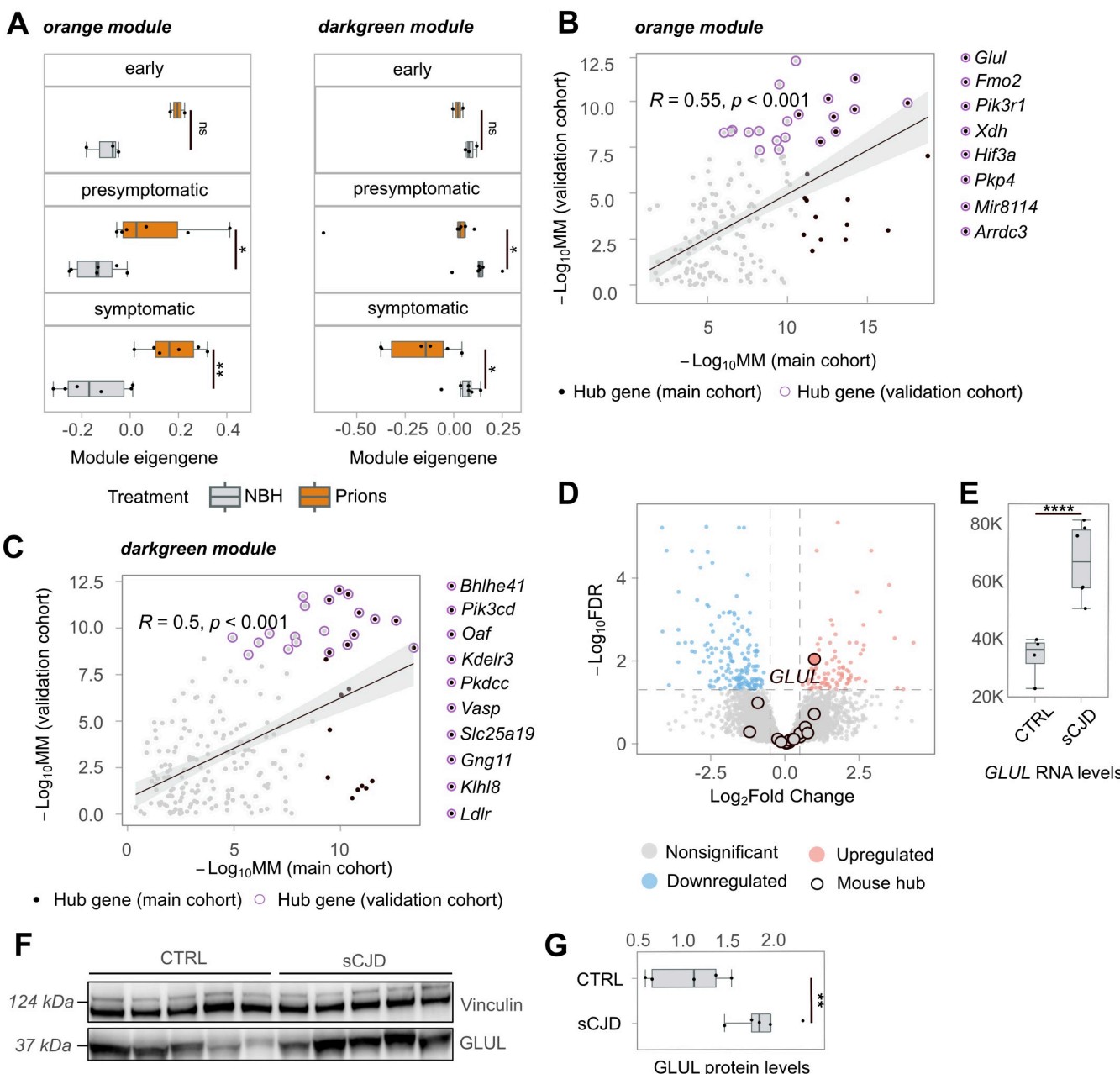

**Fig 3. Gene Co-expression and Human Validation of GLUL Upregulation. (A)** Boxplots of module eigengenes of the validation cohort for gene co-expression orange and darkgreen modules identified by WGCNA at different timestages (early, presymptomatic and symptomatic). **(B-C)** The scatter plots (Pearson's correlation and pvalue indicated with R and p, respectively) depict the relationship between genes from the orange and darkgreen modules in the main and validation cohorts. Hub genes detected in the main cohort are represented by black dots, while hub genes detected in the validation cohort are represented by purple-circled dots. The black, purple-circled dots indicate hub genes detected in both cohorts. **(D)** The volcano plot displays the results of bulk RNA sequencing analysis of skeletal muscles from patients with sCJD and their age-matched controls. Red dots represent genes that are significantly upregulated in sCJD, while blue dots represent genes that are significantly downregulated. Mouse hub genes detected in the orange and darkgreen modules are black-circled. **(E)** Boxplots with normalized GLUL transcript counts in skeletal muscles of sCJD cases and their age-matched controls. **(F)** Western blot analysis (arbitrary densitometry unit, ADU) of GLUL and Vinculin protein expression in skeletal muscle samples from sCJD cases and age-matched controls. Each lane represents a biological replicate. **(G)** Densitometry (ADU) quantification of the Western Blot in Fig 3F. Statistical significance ($^*$p < 0.05, $^{**}$p < 0.01, $^{***}$p < 0.005, $^{****}$p < 0.001) is indicated by asterisks.

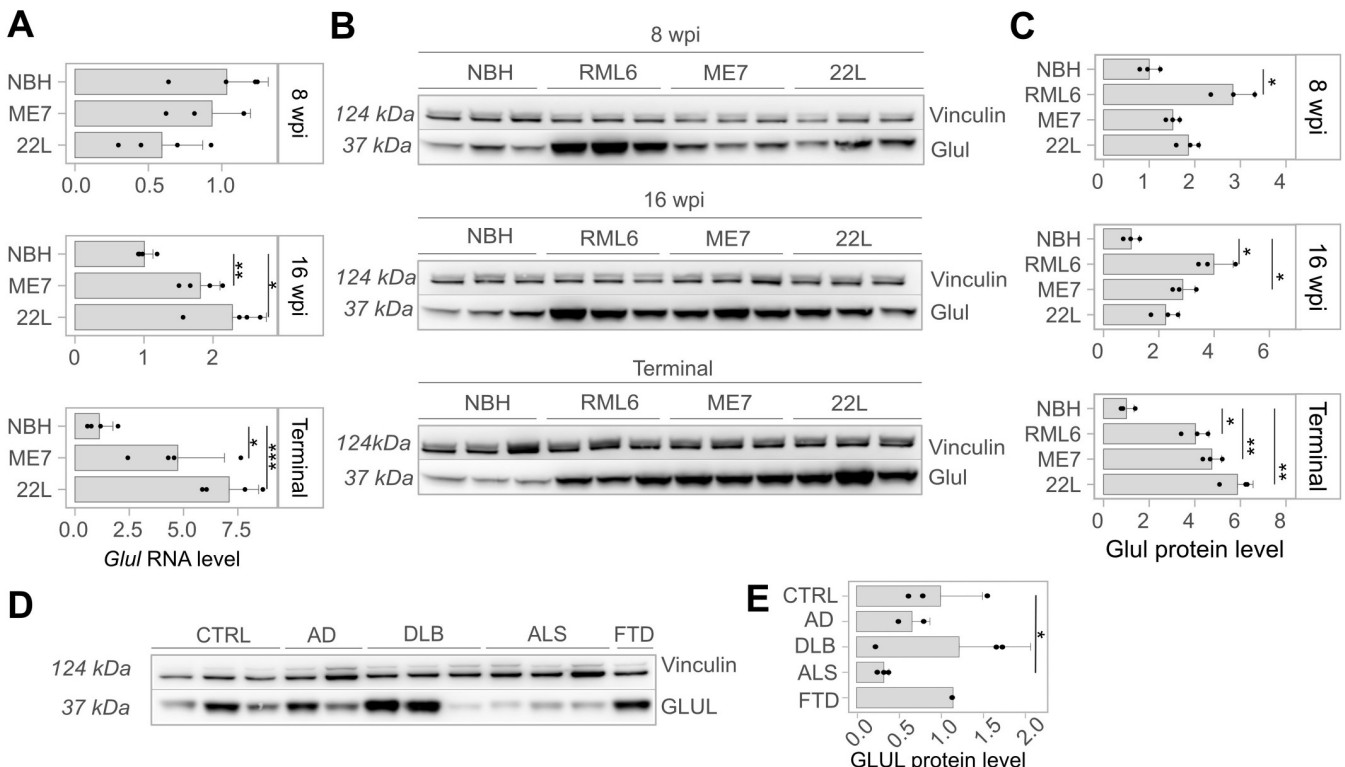

**Fig 4. Levels of Glul mRNA, protein, glutamate, and glutamine in skeletal muscle lysates at 8 and 16 weeks post-inoculation (wpi) and terminal stage of mice with prion strains RML6, ME7, and 22L, as well as related control (NBH).** In panel **(A)**, barplots display *Glul* mRNA levels normalized by GAPDH mRNA levels (derived from Ct values via RT-PCR). **(B)** Western blots of Glul and Vinculin protein levels of infected mice with different prion strains, as well as related NBH control **(C)** Densitometry (arbitrary densitometry unit, ADU) quantification of the Western Blot in Fig 4B. **(D)** Western blot of Glul and Vinculin protein levels of skeletal muscles from AD, DLB, ALS and FTD diagnosed individuals **(E)** Densitometry (ADU) quantification of the Western Blot in Fig 4D. Statistical significance (*p < 0.05, **p < 0.01, ***p < 0.005, ****p < 0.001) is indicated by asterisks.

At 8 wpi, *Glul* RNA and protein levels were not altered in both ME7 and 22L strains. At the protein level, the most substantial upregulation was found in RML6-infected animals (Fig 4A–4C). The delayed *Glul* upregulation in mice inoculated with the ME7 and 22L strains may be related to differences in disease onset, as RML6 induced disease more quickly than ME7 and 22L. *Glul* expression was significantly altered at both the RNA (S8 Table) and protein level in RML6 and ME7 strains, while only at RNA level in 22L strain at 16 wpi. In terminally sick mice, both RNA and protein level of Glul were upregulated in all strains (Fig 4A–4C). These results suggest that Glul upregulation is a universal feature across various prion diseases, highlighting its potential role in the underlying pathophysiological processes of these conditions. Such a consistent pattern of dysregulation suggests that glutamate ammonia ligase may influence disease progression and clinical manifestations in prion-affected individuals.

## GLUL upregulation is specific to prion diseases

To further investigate the disease specificity of *Glul* dysregulation, we examined its protein levels in hindlimb skeletal muscles of mouse models for ALS, AD, and DLB. Interestingly, Glul did not exhibit upregulation in these models. The confinement of Glul upregulation to prion diseases points to specific pathogenetic mechanisms that do not occur in other types of neurodegenerations (S8A and S8B Fig). To corroborate these findings, we extended our investigation to skeletal muscle necropsies from patients with familial ALS, FTD (Frontotemporal Dementia), AD, and

DLB. Again, GLUL protein levels were unaltered in AD, DLB and FTD, highlighting its distinct role in prion diseases compared to other neurodegenerative conditions. Notably, GLUL protein levels were significantly downregulated in all three individuals affected by ALS (Fig 4D–4E).

## Glutamate-glutamine biosynthesis in muscle is affected during prion disease progression

Glul catalyzes the conversion of glutamate to glutamine, whereas glutaminase carries out the opposite process. Given the increased *Glul* expression in skeletal muscle during prion disease progression, we investigated whether this metabolic pathway was affected. We measured glutaminase, glutamate and glutamine levels in muscle lysates from C57BL/6 mice inoculated with ME7, 22L and RML6 brain homogenates, as well as NBH for control. At the early disease stage, there were no significant alterations in the levels of glutamate and glutamine (S9A and S9E Fig), even in RML6-infected animals, despite the increased expression levels of Glul. This might be attributable to a compensatory effect of glutaminase upregulation (S8D and S8E Fig) which may maintain glutamate homeostasis and offset the increased expression of Glul. Notably, we observed an increase in the expression of Glul at 16 wpi (two weeks prior to the onset of clinical signs of prion disease). Despite the Glul upregulation, the balance of glutamate and glutamine was maintained (S9A and S9E Fig), as well as glutaminase levels (S8D and S8E Fig), pointing to a metabolic ability restoring the levels of glutamate consumed by Glul upregulation. At the terminal stage of the disease, the levels of glutamate were conspicuously reduced (S9A Fig), plausibly due to a significant upregulation of Glul. However, the levels of glutamine and glutaminase were unchanged (Figs 9E, S8D and S8E). Glutamate reduction was also detected in muscle necropsies of sCJD patients versus age- and sex-matched controls (S9B Fig) despite unchanged glutamine levels (S9F Fig).

We then examined the levels of glutaminase, glutamate, and glutamine in skeletal muscles of aged mouse models for AD, DLB, and ALS. Unlike in prion diseases, we found no significant alterations in glutaminase (S8A and S8C Fig), glutamate (S9C Fig), and glutamine (S9G Fig) levels in these neurodegenerative disorders. This highlights the specificity of the observed metabolic changes in prion diseases. We then evaluated glutamate and glutamine levels in skeletal muscles of human AD, DLB, ALS and FTD cases. Our results revealed a reduction in glutamate levels solely in the context of ALS, mirroring the observed trend in prion diseases (S9D and S9H Fig). However, it is important to note that while GLUL expression was downregulated in ALS, it was upregulated in prion diseases.

## Discussion

Transcriptional changes observed in the early phases of PrDs offer valuable insights into the pathogenic mechanisms at play in extraneural organs. These alterations in gene expression not only enhance our understanding of the disease's progression but also underscore the systemic nature of PrDs, revealing critical aspects of their pathology beyond the central nervous system. Here we conducted a comprehensive transcriptomic characterization of extraneural organs known to harbor prions. We hypothesized that the presence of prions in these organs could result in changes to RNA processing, or modifications to the abundance of specific transcripts. By examining multiple timestages throughout the progression of the disease in prion-inoculated mice, this investigation provides insights into the pathological processes preceding the onset of clinical symptoms. The selection of timestages was based on a recent study where authors defined disease stages based on clinically relevant EEG (electroencephalography): recordings between NBH and RML mice groups began to diverge at 10 wpi and became significantly

different at 18 wpi marking the beginning of clinical onset [32]. Other authors defined symptomatic stage as the period from 18 wpi to terminal stage based on motor function impairments [31].

Significant changes in gene expression were observed in all three organs. In the spleen, major transcriptional alterations were detected only at the terminal stage of the disease, although prions were locally present from the very beginning of the prion pathology. The early arrival of prions in the spleen may be due to the entry of a portion of the inoculum into the vascular compartment during the injection [45] combined with efficient prion replication in the follicular dendritic cells that form the scaffold of lymphoid follicles [46–50]. Such evidence indicates that, although prions accumulate in the spleen, they do not cause significant transcriptional derangements. Notably, this finding aligns with previous research showing that prion infiltration in lymphoid tissues does not lead to substantial morphological or functional alterations. Indeed, infected animals can sustain normal humoral and cell-mediated immune responses [51–53].

Conversely, blood samples displayed upregulation of numerous transcripts during the early stages of prion exposure, of which a significant fraction was related to hemostasis.

Except for *Flnb*, *Copa* and *Cog3*, we found no evidence for broad dysregulation of posttranscriptional RNA editing, in contrast to a recent report [54] but in line with our previous findings [31]. Furthermore, splicing analysis suggests that alternative splicing was largely unlinked from gene expression changes in extraneural organs.

Gene-expression changes were detected in both blood and spleen but were not consistent across all timepoints. However, the transcriptome of skeletal muscle exhibited consistent alterations throughout disease progression, even though prions were detectable only at the presymptomatic and symptomatic stages in mice infected with RML prion strain. It suggests that many of these changes were secondary consequences of prion spread in the CNS. Using WGCNA, we identified two primary gene subsets in skeletal muscle exhibiting progressive changes that became evident in the early stage of disease. Validation of these findings in an independent cohort of mice, in which sequencing libraries were prepared several months after RNA extraction from snap-frozen skeletal muscles by a different researcher, enhances the robustness of the results. This validation underscores the consistency of the findings even in the face of significant technical variability introduced by the procedures. These results have potential applications in monitoring disease progression following therapeutics administration, in conjunction with molecular and behavioral assessments for evaluating treatment efficacy [55–57].

We evaluated the relevance of our findings in humans by conducting RNA sequencing on skeletal muscle samples obtained from individuals with sCJD and control individuals without neurological impairments. While we were unable to confidently evaluate the level of preservation of mouse and human muscle gene-expression networks, we found little overlap between the joint set of hub genes identified across two mouse cohorts and human DEGs. There may be various reasons for this discrepancy, including genetic differences between mice and humans resulting in varying biological responses to similar diseases [58]. Additionally, different conditions were employed to process the human samples and decontaminate them from prions, which may have influenced the results [59]. Variations in disease stage or severity, as well as differences in the tissue types analyzed (hindlimb muscle in animals vs. psoas muscle in humans), could have also played a role in the observed differences [60]. Finally, there may be underlying differences in the disease biology between humans and mice that are not yet fully understood or characterized [61]. Despite the abovementioned limitations in translating mouse findings to human ones, we found that *GLUL*, which was a strong hub gene in the upregulated "orange" module in the mouse study, was significantly upregulated in human sCJD samples, as confirmed at the protein level by Western Blot analysis. Hence, the consistent upregulation of GLUL may be indicative of its significant role in the pathophysiology of human PrDs, potentially contributing to the progression of the disease, especially in its early

stages. GLUL is a glutamate-ammonia ligase that catalyzes the synthesis of glutamine from glutamate and ammonia in an ATP-dependent reaction. While prions were undetectable in skeletal muscles at early stages in mice infected with the RML6 prion strain, Glul upregulation was already evident, suggesting it might originate as a secondary consequence of brain infection or from other affected organs. Despite the presence of prions in the skeletal muscles of mice infected with ME7 and 22L prion strains, Glul upregulation was milder than that seen with RML6 prions, supporting the hypothesis that such upregulation is not directly caused by on-site prions in skeletal muscle.

There is some evidence that *GLUL* expression in the brain is not altered in PrDs. *Glul* mRNA levels were unchanged in the brains of mice infected with prions compared to uninfected mice [31]. Another recent study found that *GLUL* expression was also unchanged in the brains of patients with sCJD compared to healthy controls [62]. Therefore, the alterations in glutamate/glutamine metabolism in prion-infected brains [63] that were proposed to contribute to neurodegeneration and cognitive dysfunctions, are unlikely to stem from any changes in brain-resident GLUL. Instead, *Glul* expression was consistently upregulated in the skeletal muscles of animals throughout the disease, and this finding was present after infection with multiple prion strains. This suggests that *GLUL* may also be upregulated before the clinical onset of human PrDs. The identification of a common pathological phenotype among these diseases would be a significant finding, shedding light on the underlying mechanisms specific to these conditions at early stages. It is important to note that, despite the limited availability of skeletal muscle specimens from patients with neurodegenerative diseases, this phenomenon appears to be unique to prion diseases and does not occur in other common neurodegenerative disorders such as ALS, AD, or DLB. Interestingly, GLUL protein levels were consistently found to be downregulated in human ALS. This distinct pattern underscores the unique role of *GLUL* in the pathophysiology of PrDs, highlighting their differentiation from other neurodegenerative conditions in terms of molecular and cellular mechanisms. We hypothesized that the upregulation of GLUL protein in skeletal muscle may be linked to a metabolic dysfunction in the glutamate-glutamine pathway. Prior research had shown the direct influence of glutamine levels on the expression of *GLUL* in skeletal muscle cells [64]. Building upon this finding, it is plausible to speculate that a potential systemic deficit in glutamine levels might contribute to the observed upregulation of *GLUL* in skeletal muscle. Skeletal muscle exerts a pivotal role in glutamine storage, production, and release into the bloodstream. This hypothesis gains further support from the significant reduction in glutamate levels observed, which could be attributed to the heightened activity of GLUL in response to altered glutamine availability. This was unexpected as previous studies had reported increased levels of glutamate and glutamine in the brain and cerebrospinal fluid of patients with neurodegenerative diseases, including PrDs [63]. The regulation of glutamate and glutamine metabolism in skeletal muscle differs from that in the brain, and the upregulation of *GLUL* in skeletal muscle may have a distinct role in the pathology of PrDs In contrast, the absence of such *GLUL* upregulation in other NDDs might signify a distinct metabolic response. The lack of compensatory *GLUL* expression could contribute to the sustained alterations in glutamine and glutamate levels seen in those conditions. Therefore, the specific association between *GLUL* upregulation and prion diseases suggests a unique interplay between glutamine/glutamate metabolism and disease progression, setting PrDs apart from other NDDs.

## Materials and methods

### Ethics statement

Animal experiments were approved by the Veterinary Office of the Canton Zurich (permit numbers ZH41/2012, ZH90/2013, ZH040/15, ZH243/15) and carried out in compliance with

the Swiss Animal Protection Law. In Switzerland, permit numbers are equivalent to approval numbers. This project is a follow-up to a previous study published in PLOS Pathogens, where we used the same specification format for the animal permits (you can find the article here). The study was approved by the Federal Office for Food Safety and Veterinary Medicine (Veterinäramt Zürich, Zollstrasse 20, 8090 Zürich) by Dr. Simone Gilg. The animals used in this study are the same as those in the aforementioned publication. Animal discomfort and suffering was minimized as much as possible and individual housing was avoided.

We obtained sCJD anonymized skeletal muscle samples from an approved study sanctioned by the Cantonal Ethics Committee of the Canton of Zurich under approval number #2019–01479 and a written informed consent was received prior to participation.

The experimental protocols involving GLUL's specificity validation test on other neurodegenerative diseases by using human participants adhered strictly to the guidelines set by French regulations. Prior to participation, comprehensive written informed consent was diligently obtained from all individuals, including those who underwent skeletal muscle necropsies for Familial Amyotrophic Lateral Sclerosis (Familial SLA), Fronto-Temporal Dementia with ALS (DFT-SLA), and pure cases of Alzheimer's Disease (AD) and Dementia with Lewy Body (LBD). Human biological samples and associated data were obtained from Tissu-Tumorothèque Est (CRB-HCL, Hospices Civils de Lyon Biobank, BB-0033-00046).

## Animals used in the study

Prion-inoculated and control-injected mice were regularly monitored for the development of clinical signs, according to well-established procedures using humane termination criteria. Intracerebral injections and transcardiac perfusions were performed in deeply anesthetized mice. Habituation periods before the experiment began were included. Male C57BL/6J mice were obtained from Charles River, Germany. Mice were housed in a conventional sanitary facility and monitored for the presence of all viral, parasitic, and bacterial species listed in the Federation of European Laboratory Animal Associations (FELASA). The facility was tested positive for Murine Norovirus and Helicobacter spp. The mice were housed in IVC type II long cages and up to five animals were housed in the same cage which were staffed with individual apartments. Mice had unrestricted access to sterilized drinking water and were fed *ad libitum* a pelleted mouse diet. The light/dark cycle consisted of 12/12 h with artificial light. The temperature in the room was 21 ± 1˚C with a relative humidity of 50 ± 5%. Air pressure was regulated at 50 Pa, with 15 complete changes of filtered air per hour by a HEPA filter.

## Prion inoculations and processing of tissue samples

Animals, prion inoculation and necropsy procedures are identical to those described in [31]. C57BL/6J male mice were inoculated in the right hemisphere with either 30 μl of passage 6 of Rocky Mountain Laboratory (RML6), or 22L, or ME7 strain mouse-adapted scrapie prions containing 9.02 $LD_{50}$ of infectious units per ml in 10% w/v homogenate. Non-infectious brain homogenate (NBH) from CD1 mice was used as a negative control. Mice were assigned randomly to experimental groups. Animals were monitored at least thrice per week, after the clinical onset of PrDs, they were monitored daily, and prion-inoculated mice were terminated upon evident signs of terminal disease. NBH-inoculated mice were sacrificed 13 days after the termination of the last prion-inoculated mice. Whole blood, spleen and muscle were dissected, snap-frozen in liquid $N_2$ and stored at -80˚C prior to sequence library generation.

## Processing of AD, DLB and ALS tissue samples

Double-transgenic APP/PS1 mice (n = 3; 8 months of age) were used as AD mouse models from which we have collected hindlimb skeletal muscles. For the DLB mouse model, hindlimb skeletal muscles of transgenic A53T synuclein mutant mice lines were kindly provided by Dr. Noain Daniela's group (Department of Neurology, University Hospital Zurich; n = 2, 8 months of age) and by Dr. Ruiqing Ni's group (Institute for Biomedical Engineering, University Hospital Zurich; n = 1, 8 months of age). Non-transgenic C57BL/6J male littermates (n = 3, 8 month old) were used as controls. All mice were housed under a 12-hour light/12-hour dark schedule and had free access to food and water. All animals were euthanized by pentobarbital injection. Skeletal muscles were dissected, snap frozen in liquid $N_2$ and stored at -80˚C prior to western blot and biochemical analyses.

Differently, skeletal muscle lysates obtained from both wild-type and SOD1$^{G93A}$ transgenic mouse models of ALS were generously provided by Prof. Dr. Musarò, the Principal Investigator leading the neuromuscular research group at the Sapienza University of Rome. To provide a concise overview, wild-type C57BL/6 (WT) and transgenic SOD1$^{G93A}$ mice were utilized for our investigation. The mice were sacrificed at 130–140 days of age, a time point closely aligned with the spontaneous mortality of SOD1$^{G93A}$ mice. Euthanasia was conducted through cervical dislocation to ensure minimal suffering. Immediately following the humane sacrifice, muscle samples were excised for subsequent analysis, with one muscle specimen collected from each animal for testing purposes. Tissue lysates were prepared according to [65].

## Preparation of RNA libraries for Mouse sequencing

RNA was extracted from snap-frozen organs by means of the RNeasy Plus Universal Kit (QIAGEN). The quantity and quality of RNA were analyzed with Qubit 1.0 Fluorometer (Life Technologies) and Bioanalyzer 2100 (Agilent Technologies), respectively. For library preparation, we used TruSeq RNA Sample Prep kit v2 (Illumina). We performed poly-A enrichment on 1 μg of total RNA per sample, which was then reverse transcribed into double-stranded cDNA followed by ligation of TruSeq adapters. Sequencing fragments containing TruSeq adapters at both termini were enriched by PCR. Quantity and quality of enriched libraries were analyzed using Qubit (1.0) Fluorometer and Caliper GX LabChip GX (Caliper Life Sciences), which showed a smear corresponding to a mean fragment size of around 260 bp. Libraries were then normalized to 10 nM in Tris-Cl 10 mM, pH 8.5, with 0.1% (v/v) Tween 20. Cluster generation was performed with the TruSeq PE Cluster kit v4-cBot-HS (Illumina), using 2 pM of pooled normalized libraries on the cBOT. Sequencing was performed on Illumina HiSeq 4000 paired-end at 2 × 126 bp using the TruSeq SBS kit v4-HS (Illumina).

## Post-trascriptional changes analysis

Adenosine-to-inosine editing index (AEI) was calculated as previously published [33]. Herein, raw fastq reads were uniquely aligned to a murine mm10 reference genome using STAR v2.7.3 with the filter outFilterMultimapNmax = 1. RNAEditingIndexer (https://github.com/a2iEditing/RNAEditingIndexer) was used to calculate per-sample AEI.

We identified gene-specific RNA editing based on a recently published list of high-confidence targets of *Adar* [34] as follows. RediToolsKnown.py from REDItools [66] was applied on uniquely aligned samples as mentioned above. This yielded per-site lists of A-to-I editing on which we applied the following thresholds: (a) a minimum of 3 alternative reads per site per sample (b) a minimal editing frequency of 1% per site (c) criteria a) and b) are fulfilled in at least floor(2/3 * n) biological replicates, n is total number of biological replicates per group (d) transcripts of site present in at least 2 biological control replicates. Multiple testing of sites

passing above-mentioned thresholds was performed using REDIT (https://github.com/gxiaolab/REDITs) and adjusted for false discovery rate (FDR) according to Benjamini-Hochberg, we considered sites with an FDR < 0.05 to be significantly edited.

For alternative splicing, SGSeq R package [67] was employed to find splicing events characterized by two or more splice variants. Exons and splice junction predictions were obtained from BAM filesPrediction of exons and spliced junctions was first made for each sample individually. Then the predictions for all samples were merged and we obtained a common set of transcript features. Overlapping exons were disjoint into non-overlapping exon bins and a genome-wide splice graph was compiled based on splice junctions and exon bins. A single value for each variant was produced by adding up the 5' and 3' counts, or, if these represented the same transcript features, by considering the unique value. These counts were then fed to DEXSeq [68]. We analyzed differential usage of variants across a single event, in-stead of quantifying differential usage of exons across a single gene. We retained only variants with at least five counts in at least three samples (of any condition). After filtering, the events associated with a single variant were discarded. Differential analysis was then performed implementing a sample+exon+condition:exon model in DEXSeq. Differentially expressed isoforms were defined as isoforms changing with FDR < 0.05. In the case of differentially used splice variants in muscle on 12 wpi, this dataset was considered as an outlier and hence excluded due to excessively reported splice variants (1,788 events compared to 5 or less on all other time-points and extraneural organs).

## Patient samples

Human skeletal muscle samples of psoas major muscle were collected from patients with a clinical suspicion of Creutzfeldt-Jakob's disease and submitted for an autopsy to the Swiss National Reference Center for Prion Disease between 2004–2011. A detailed description of samples used for RNA extraction and sequencing is given in S6 Table. Sporadic CJD was diagnosed according to criteria described previously [69].

## Preparation of RNA libraries for sequencing of human tissues

Firstly, CJD bulk tissues were lysed in TE buffer with the anionic detergent sodium dodecyl sulphate (SDS) and digested at 50˚C with 2 mg / ml$^{-1}$ Proteinase K (PK) for 2 hours to eliminate solids and release DNA/RNA from proteins. Although prions are well-known for their relative resistance to PK digestion, prion infectivity largely depends on PK-sensitive oligomers. Indeed, prolonged PK digestion reduces prion titers by a factor of $>10^6$, but residual PK-resistant material may still be infectious. In a second step, TRIzol reagent solution was added to the lysate (it contains Gdn-SCN and phenol, which inactivate RNases and disaggregate prions) and kept overnight at 4˚C. Gdn-SCN is a chaotropic salt which rapidly denatures proteins and abolishes the infectivity of prion inoculum. At high concentrations, guanidine salts disaggregate PK-resistant PrP$^{Sc}$ fibrils, eliminate PK resistance and abolish PrP$^{Sc}$ conversion, meaning that any PK-resistant material that survived the digestion step would be expected to be inactivated at this stage of the protocol. 0.2 ml of ultrapure phenol:chloroform:isoamyl alcohol (Thermo Fischer Scientific) was added to the samples, followed by strong shaking and incubation at room temperature for 5 mins. Centrifugation step at 12,000 x g for 15 min at 4˚C generated two phases. The aqueous upper phase was transferred to a fresh tube; 0.5 ml of isopropanol and 1 μl of Glycoblue Coprecipitant (Thermo Fisher Scientific) were added. Next, RNA was pelleted for 20 min at 12,000 x g at 4˚C and washed twice with 75% ethanol. The RNA pellet was dissolved at 55˚C in 20 μl of free nuclease water.

The quantity and quality of RNA were analyzed with Qubit 1.0 Fluorometer (Life Technologies) and Tapestation 4200 (Agilent Technologies), respectively. The TruSeq stranded RNA protocol (Illumina) was employed for library preparation. In brief, 1 μg of total RNA per sample was poly-A enriched, reverse transcribed into double-stranded cDNA and then ligated with TruSeq adapters. PCR was performed to selectively enrich for fragments containing TruSeq adapters at both ends. The quantity and quality of enriched libraries were analyzed using Qubit (1.0) Fluorometer and Tapestation 4200. The resulting product is a smear with a mean fragment size of approximately 260 bp. Libraries were then normalized to 10 nM in Tris-Cl 10 mM, pH 8.5, with 0.1% (vol/vol) Tween 20. Cluster generation was performed with the TruSeq PE Cluster kit v4-cBot-HS (Illumina), using 2 pM of pooled normalized libraries on the cBOT. Sequencing was performed on Illumina HiSeq 4000 paired-end at $2 \times 126$ bp using the TruSeq SBS kit v4-HS (Illumina).

Despite profound RNA degradation, we speculated that moderate RNA degradation might preserve biological information. We then decided to set a RIN $\geq 3$ as a minimal threshold for human muscle tissue. 12 out of 28 initially collected samples passed the RIN threshold (S4A Fig and S6 Table) and were further processed. 2 out of 12 samples were subsequently removed because of a high cluster condition variance (upon the quality control of the sequencing) resulting in a final sample size of n = 10 for downstream analysis.

## Differential gene expression

We used FASTQC and parallel [70] algorithms for quality control of raw sequencing reads. We clipped low-quality ends as follows: 5' = 3 bases, 3' = 10 bases. Reads were aligned to mouse mm10 and human GRCh38.p13 reference genome, and transcriptome using STAR v2.3.0e_r291 [71] on cloud computing solution SUSHI of the Functional Genomics Center of Zurich [72]. DESeq2 [43] was used to detect differentially expressed genes based on the following thresholds: (a) $|\log_2$-fold change$| > 0.5$ (b) FDR $< 0.05$. Genes with less than 10 counts in total were excluded. Sex was included as a covariate in the formula for analyzing human samples with DESeq2. Gene ontology analysis was performed using clusterProfiler for R [73].

## WGCNA

WGCNA was performed using the WGCNA R package [74]. Outlier genes were identified and removed using the *goodSamplesGenes()* function. Additionally, genes with fewer than 10 counts in over 50% of samples were filtered out. Raw count data was normalized using the variance stabilizing transformation provided by the DESeq2 R package. An adjacency matrix was generated using the *adjacency()* function with default parameters. To meet the criteria for a scale-free network, a soft threshold of 4 was uniformly applied to all networks. Adjacency matrix was transformed into a Topological Overlap Matrix (TOM). Average linkage hierarchical clustering was performed on a dissimilarity matrix (1 –TOM) and subsequently, modules of co-expressed genes were identified using the dynamic cut tree algorithm (*cuttreeDynamic function()*), with a minimum cluster size set to 30. Similar modules were merged based on their module eigengene (ME) correlation. To assess the significance of differences in ME values between tested conditions, a Mann-Whitney U test was conducted. To identify genes with the highest connectivity within modules, Module Membership (MM) was computed as the Pearson correlation coefficient (p-value) between individual gene expression levels and the ME. For genes in the modules of interest, we calculated a gene significance score based on the p-values calculated with DESeq2 for each timestage. Specifically we combined p-values for different time stages using the *combineParallelPValues()* function from the metapod R package with the method argument set to "stouffer" [75]. Negative log base 10 of the combined p-value

represents the gene significance score. The preservation of mouse muscle co-expression network was tested using *modulePreservation()* function from the WGCNA package, using the network from the main cohort as a reference.

## Prions detection in the spleen

50 mg of spleen tissue was homogenized in 500 μl lysis buffer (0.5% wt/vol sodium deoxycholate and 0.5% vol/vol Nonidet P-40 in PBS) at 5'000 rpm for 5 minutes by using a Precellys24 Sample Homogenizer (LABGENE Scientific SA, BER300P24) and incubated on ice for 20 minutes. The cleared lysates were obtained by centrifugation at 1'000 rcf for 6 minutes in an Eppendorf 5417 R tabletop centrifuge. For PK-WB, the concentration of whole protein was determined using a BCA assay (Thermo Scientific). 100 μg of total protein was digested with 20 μg ml$^{-1}$ proteinase K for 30 min at 37 °C, then mixed with western blotting loading buffer and boiled for 10 min at 95 °C. The following primary antibodies were used: mouse monoclonal antibody against vinculin (1,5000, Abcam, ab129002); mouse monoclonal antibody against PrP (POM1, 1:5,000, homemade) for prions detection; mouse monoclonal antibody against PrP (POM2, 1:5,000, homemade) for PrP$^C$ detection.

## Prions detection with NaPTA protocol in the skeletal muscle

To detect prions in skeletal muscles, we performed sodium phosphotungstic acid (NaPTA) enrichment [42]. NaPTA binds and precipitates PrP$^{Sc}$ in the presence of MgCl$_2$, removing contaminants and concentrating PrP$^{Sc}$. For this protocol, 40 mg of tissue were lysed in 400 μl of 2% sarcosyl-PBS by using Precellys24 Sample Homogenizer (LABGENE Scientific SA, BER300P24) twice, at 5'000 rpm for 5 minutes. To remove gross debris, samples were centrifuged at 80g for 1 minute. Samples were next incubated at 37°C while shaking at 1500 rpm for 30 minutes. Next, 50U/ml of benzonase (to degrade DNA contaminants) and 1mM of MgCl$_2$ were added and incubated at 37°C while shaking at 1500 rpm for 30 minutes. Then, 4% NaPTA and 130 mM MgCl2 were added, resulting in a final NaPTA concentration of 0.3%. The samples were incubated at 37°C while shaking at 1500 rpm for 30 minutes, followed by centrifugation at 16,000 g for 45 minutes to precipitate PrP$^{Sc}$. The resulting pellets were resuspended in 30 μl of 0.1% sarcosyl. 20 μl were next used for the PK-WB and digested with 20 μg ml$^{-1}$ proteinase K for 1 hour at 37 °C. The enzymatic reaction was blocked by stop-buffer (1% (w/v) SDS; 25 mM Tris/HCl, pH 7.4; 2,5% (v/v) β-mercaptoethanol; 1.5% (w/v) sucrose; 0.02% (w/v) brome-phenol-blue). At this point, samples were boiled for 10 min prior to Western blot analysis. NBH and RML6 brain tissue underwent the same NaPTA procedure and used as negative and positive controls. Mouse monoclonal antibody against PrP (POM1, 1:5,000, homemade) was used for prions detection.

## Western blot analysis

To prepare the samples, 1 ml of cell-lysis buffer (20 mM Hepes-KOH, pH 7.4, 150 mM KCl, 5 mM MgCl2, 1% IGEPAL) supplemented with protease inhibitor cocktail (Roche 11873580001) was added to the lysed samples. They were then homogenized twice at 5'000 rpm for 15 seconds using a Precellys24 Sample Homogenizer (LABGENE Scientific SA, BER300P24) and incubated on ice for 20 minutes. The cleared lysates were obtained by centrifugation at 2'000 rcf, 4° C for 10 minutes in an Eppendorf 5417 R tabletop centrifuge. The concentration of whole protein was determined using a BCA assay (Thermo Scientific). The samples were boiled in 4 x LDS (Invitrogen) containing 10 mM DTT at 95°C for 5 minutes. 15 μg of total protein per sample were loaded onto a 4–12% Novex Bis-Tris Gel (Invitrogen) gradient for electrophoresis at 80 V for 15 minutes, followed by constant voltage of 150 V. The

PVDF or Nitrocellulose membranes were blocked with 5% Sureblock (LubioScience) in PBS-T (PBS + 0.2% Tween-20) for 1 hour at room temperature. Membranes were then cut into three parts according to the molecular weight. The membrane was divided into three segments for targeted antibody incubation: The upper portion was treated with anti-Vinculin (1:5000, Abcam, ab129002), the middle section with anti-Glutaminase (1:3000, Abcam, ab202027), and the lower segment with anti-Glutamine synthetase (1,2000, Abcam, ab176562). This antibody incubation was carried out in PBS-T supplemented with 1% Sureblock, and the membranes were left overnight at 4˚C to facilitate optimal binding. They were washed thrice with PBS-T for 10 minutes. The membranes were incubated with secondary antibodies conjugated to horseradish peroxidase (HRP-tagged goat anti-rabbit IgG (H+L), 1:3000, 111.035.045, Jackson ImmunoResearch) for 1 hour at room temperature. The membranes were washed thrice with PBS-T for 10 minutes and developed using a Classico chemiluminescence substrate system (Millipore). The signal was detected using a LAS-3000 Luminescent Image Analyzer (Fujifilm) and analyzed with ImageJ software.

### RT-QuiC protocol

RT-QuIC assays were conducted as previously described [76]. In brief, 10% wt/vol spleen homogenates (0.5% wt/vol sodium deoxycholate and 0.5% vol/vol Nonidet P-40 in PBS) were diluted 2000-fold and used as seeds for the RT-QuIC reactions. For skeletal muscle analysis, 2 μl of the NaPTA prion precipitation product from a 10% wt/vol sample, resuspended in 30 μl of 0.1% sarcosyl-PBS, were used. Recombinant hamster PrP (HaPrP) served as the monomeric substrate for RT-QuIC conversion. The reactions contained HaPrP substrate protein at a final concentration of 0.1 mg/ml in PBS (pH 7.4), 170 mM NaCl, 10 μM EDTA, and 10 μM Thioflavin T, with 2 μl of diluted brain homogenates added to a total volume of 100 μl. NBH and RMl6 brain homogenates were used as negative and positive, respectively.

The RT-QuIC reactions consisted of 100 hours with intermittent shaking cycles set at 42˚C: 90 seconds of shaking at 900 rpm in double orbital mode, followed by 30 seconds of rest, using a FLUOstar Omega microplate reader (BMG Labtech). Thioflavin T fluorescence was measured every 15 minutes to monitor aggregate formation (450 nm excitation, 480 nm emission; bottom read mode).

### Biochemical analysis

The concentration of glutamine and glutamate in skeletal muscle was measured using the Merck Glutamine Assay Kit (Catalog Number MAK438). To prepare the lysates, a total of 600 μg of total protein was utilized. In each standard and sample well, 80 μL of working reagent was added to determine the glutamine concentration. To measure glutamate concentration, samples were also prepared with 80 μL of blank working reagent. The 96-well plate was incubated at 37˚C for 40 minutes. Absorbance values were recorded at 450 nm using a microplate reader. The concentrations of glutamine and glutamate were determined by comparing the absorbance of the samples to a standard curve generated from known concentrations of glutamine and glutamate. The results were expressed as μM/ml.

### Statistical analysis

For Fig 3E, normalized raw counts for the GLUL gene in control and sCJD patients were analyzed using the DESeq2 package, with related false discovery rate (FDR) calculations. In Fig 3G, Western blot densitometry data were analyzed using the Mann-Whitney U test, resulting in a p-value of 0.01072. For statistical analyses in Fig 4A, Mann-Whitney U test was used. In Figs 4C, S8B, S8C and S8E), we applied the t-test (as the standard deviations were consistent)

with Bonferroni correction to account for multiple comparisons. In S9 Fig, the Mann-Whitney U test was used due to its non-parametric nature, which is suitable for data that do not assume a normal distribution.

The choice of DESeq2 for RNA-seq data, Mann-Whitney U for non-normally distributed data, and t-tests with Bonferroni correction for normally distributed data ensures that our analyses are appropriately tailored to the characteristics of the data, providing reliable and valid results.

## Supporting information

**S1 Fig. Early Changes in Blood Associated with Hemostasis and Wound Healing Terms. (A)** The 270 overlapped, upregulated, blood-derived DEGs at 4wpi and 8 wpi are associated with specific Gene Ontology (GO) terms. The Randarplot displays the results of the GO over-representation analysis by Biological Process (BP) ontology class. **(B)** Expression patterns (z-score based) of genes related to hemostasis process in blood at 4 and 8 wpi.
(TIF)

**S2 Fig. Posttranscriptional Modification During Prion Disease Progression in Extraneural Organs.** Percentage of alternative splicing events (AEI) in blood **(A)**, muscle **(B)**, and spleen **(C)** samples of prion-infected (RML6) and control (NBH) mice at various time points post-inoculation. Ordinate: AEI percentage; abscissa: weeks post-inoculation. **(D)** Significantly recoded gene transcripts *Flnb* and *Copa* genes in spleen at 16 wpi. Reduced A-to-I editing of *Cog3* transcripts in blood at terminal stage. **(E)** Bars show annotation and related number of splice variants in both extraneural organs and brains computed from previously reported data [31].
(TIF)

**S3 Fig. Qualitative Detection of Prions in the Spleen During Disease Progression. (A)** RT-QuIC reactions on spleen homogenates from prion-inoculated mice (RML labeled) and related control (NBH labeled) sacrificed at specific time points. Each sample was tested in quadruplicate, and each plot represents data from biological replicates (n = 3, red, yellow and purple). Ordinate: fluorescent intensity (relative fluorescence units normalized by minimum and maximum value from NBH and RML from the same timepoint). Biological replicates were considered prion positive if 3 out of 4 technical replicates tested positive. **(B)** Western Blot analysis of spleen before (lane above: PrP Undigested) and after (lane below: PrP Digested) PK treatment. Undigested blot was probed with anti-PrP antibody (POM2). Digested blot was probed with a different anti-PrP antibody (POM1). Brain homogenates were used as controls.
(TIF)

**S4 Fig. Qualitative Detection of Prions in NaPTA-enriched Skeletal Muscle at Different Timepoints over Disease Progression. (A)** RT-QuIC reactions performed on skeletal muscle derived, NaPTA enriched homogenates from prion-inoculated mice (RML6, ME7 and 22L labeled) and related control (NBH labeled) sacrificed at specific timepoints. Each sample was tested in quadruplicate, and each plot represents data from individual biological replicates (n = 3; red, yellow and purple). Fluorescent intensity on the y axis stands for relative fluorescence units normalized by minimum and maximum value (from control and condition) to obtain percentage. Biological replicates are considered prion positive if 3 out of 4 technical replicates test positive. **(B)** Western Blot analysis of skeletal muscle after PK treatment. Digested blot is probed with anti-PrP antibody (POM1). Mice infected with different prion strain (RML6, ME7 and 22L) are shown with appropriate segment above the blot. Brain homogenates were used as controls.
(TIF)

**S5 Fig. Modul*e* Eigengene Significance Heatmap for Different Organs and Time Stages.**
Each heatmap illustrates module eigengene significance at three timestages (early, pre-symp-
tomatic, and symptomatic) derived from the comparison between NBH and RML6 inoculated
mice from **(A)** whole blood, **(B)** spleen, and **(C)** skeletal muscles necropsies. Each row repre-
sents a specific module, while columns correspond to individual timestages. Statistical signifi-
cance (*p < 0.05, **p < 0.01, ***p < 0.005, ****p < 0.001) is indicated by asterisks.
(TIF)

**S6 Fig. Module Preservation, and Module Eigengene of validation cohort. (A)** Scatter plot
of the Z-summary module preservation statistic and the sizes of modules in muscle co-expres-
sion network. The modules with Z-summary > 1.96 were interpreted as preserved. **(B)** The
heatmap illustrates module eigengene significance at three timestages (early, pre-symptomatic
and symptomatic) derived from the comparison between NBH and RML6 inoculated mice.
Each row represents a specific module, while columns correspond to individual timestages.
Statistical significance (*p < 0.05, **p < 0.01, ***p < 0.005, ****p < 0.001) is indicated by
asterisks.
(TIF)

**S7 Fig. Comprehensive Assessment of Skeletal Muscle RNA Quality and Expression Pat-
terns in sCJD Patients and Controls. (A)** Agilent Bioanalyzer gel image depicting total RNA
samples extracted from skeletal muscle tissues of both sCJD patients and control subjects**.** The
image showcases the RNA quality assessment using the Ribosomal Integrity Number (RIN)
scores, displayed alongside the respective samples. **(B)** Principal Component Analysis (PCA)
plot illustrating the segregation of gene expression profiles in skeletal muscle samples between
individuals with sCJD and non-sCJD controls. Each data point represents a distinct sample,
with colors corresponding to two sample conditions**. (C)** Heatmap illustrating the variation in
gene expression between individuals with sCJD and non-sCJD controls. Each row corresponds
to a differentially expressed gene, while each column represents an individual subject from
either the sCJD or control group. **(D)** $Log_2$ fold change of *Glul* transcript derived from the
comparison between RML6-infected and NBH treated animals at different analyzed time-
points in skeletal muscle.
(TIF)

**S8 Fig. Multi-level Characterization of GLUL in Prion-Infected Mice and Neurodegenera-
tive Disease Models, as well as Related Human Cases.** In panel **(A)** Western blots of Glul,
Glutaminase and Vinculin protein levels, of mouse models for AD, DLB and ALS, as well as
related control (C57BL/6J). This Western blot presents analysis of two distinct control groups
as obtained directly from different collaborators. **(B)** Densitometry data (ADU) quantification
of Glul (normalized with related Vinculin) from the Western blot in **S8A Fig**. **(C)** Densitome-
try data (ADU) quantification of Glutaminase (normalized with related Vinculin) from the
Western blot in **S8A**. Each lane in the Western Blots represents a biological replicate. Statistical
significance (*p < 0.05, **p < 0.01, ***p < 0.005, ****p < 0.001) is indicated by asterisks. (**D**)
Western blot of Glutaminase and Vinculin protein of mice inoculated with prion strains
RML6, ME7, and 22L, as well as related control (NBH). **(E)** Densitometry (arbitrary densitom-
etry unit, ADU) quantification of the Western blot in **S8D Fig**.
(TIF)

**S9 Fig. Glutamate/Glutamine Profiles in Prion-Infected Mice and Neurodegenerative Dis-
ease Models, as well as Related Human Cases (A)** Glutamate concentrations (nmol/mg) in
skeletal muscle lysates of **(A)** mice inoculated with prion strains RML6, ME7, and 22L, as well

as related control (NBH) at different timepoints, **(B)** of sCJD and non-sCJD control patients **(C)** mouse models of AD, DLB and ALS, as well as related control (C57BL/6J) and **(D)** human cases of AD, DLB, ALS and FTD. Glutamine concentrations (nmol/mg) in skeletal muscle lysates of **(E)** mice inoculated with prion strains RML6, ME7, and 22L, as well as related control (NBH) at different timepoints, **(F)** of sCJD and non-sCJD control patients **(G)** mouse models of AD, DLB and ALS, as well as related control (C57BL/6J) and **(H)** human cases of AD, DLB, ALS and FTD. Each dot in the graphs represents a biological replicate. Statistical significance (*p < 0.05, **p < 0.01, ***p < 0.005, ****p < 0.001) is indicated by asterisks.
(TIF)

**S10 Fig. Western blots of undigested and PK-digested samples were run independently.** The undigested Western blots were sectioned into two parts, each stained with a different antibody: anti-Vinculin and anti-PrP. Samples represent spleen of infected mice with RML6 prion strain and related NBH control sacrificed at **(A)** 8 wpi, **(B)** 16 wpi and **(C)** terminal stage.
(TIF)

**S11 Fig. Western blots of NaPTA-treated, PK-digested samples were run independently.** The Western blots were stained with anti-PrP antibody. Samples represent skeletal muscle of infected mice with different prion strains (RML6, ME7 and 22L) and related NBH control sacrificed at **(A)** 8 wpi, **(B)** 16 wpi and **(C)** terminal stage.
(TIF)

**S12 Fig.** (A) Co-staining of anti-Vinculin and anti-GLUL antibodies on a single blot. (B) Western blot strips were cut from the same gel, divided into three parts for staining with different antibodies (anti-Vinculin and anti-GLUL). The middle section was not used in this paper. (C-D) Western blot strips were cut from the same gel, divided into three parts for staining with different antibodies: anti-Vinculin, anti-Glutaminase and anti-Glul.
(TIF)

**S13 Fig. Western blot strips were cut from the same gel, divided into three parts for staining with different antibodies: anti-Vinculin, anti-Glutaminase and anti-Glul.** Samples represent skeletal muscle of infected mice with different prion strains (RML6, ME7 and 22L) and related NBH control sacrificed at **(A)** 8 wpi, **(B)** 16 wpi and **(C)** terminal stage. In the (C) panel, the strip intended for Glutaminase protein staining was cut, stained, and acquired after the Glul staining was completed.
(TIF)

**S1 Table. Overlapping blood-derived DEGs shared between 4 and 8 wpi in PrDs.**
(XLSX)

**S2 Table. Gene Ontology terms of the overlapping blood-derived DEGs shared between 4 and 8 wpi in PrDs.**
(XLSX)

**S3 Table. Gene expression changes in prion-infected mice across different time points and organs.** The table lists genes that were either upregulated or downregulated in skeletal muscle, spleen, and blood at various time intervals during prion disease progression. The direction of regulation (up or down) is indicated for each gene, highlighting significant transcriptional changes associated with prion infection in specific organs.
(XLSX)

**S4 Table. Differential splicing events in various organs of prion-infected mice at different timepoints over the disease progression.**
(XLSX)

**S5 Table. Gene modules associated with prion disease in blood, muscle, and spleen tissues.** This table categorizes genes into specific modules based on their expression patterns across different tissues (blood, muscle, and spleen) in prion-infected mice.
(XLSX)

**S6 Table. Information on the types of sCJD cases analyzed, including genotype, strain type, and age.** This table further provides the reasons of biosamples exclusion from the final bulk RNA sequencing and downstream analysis.
(XLSX)

**S7 Table. DEGs in skeletal muscle from sCJD patients compared to age-matched controls.**
(XLSX)

**S8 Table. qPCR primers for *Glul* mRNA transcript detection in skeletal muscle in prion-infected mice, and related controls.**
(XLSX)

**S1 Data Values. Table supporting data values for all data presented in graphical form.**
(XLSX)

## Acknowledgments

The authors are grateful to Martina Cerisoli, Nicola Conneely, Andrea Armani, Marigona Imeri, and Mirzet Delic for support, assistance in laboratory investigations and animal husbandry, and also Med. Dr. Regina Reimann for providing sCJD skeletal muscles biopsies. The authors acknowledge the Functional Genomics Center Zurich of the ETH Zurich, and Next Generation Sequencing Platform of University of Bern for preparing sequencing libraries, RNA sequencing, quality control and technical support of mouse and human studies, Prof. Eli Eisenberg (Raymond and Beverly Sackler School of Physics and Astronomy and Sagol School of Neuroscience, Tel Aviv University, Tel Aviv, Israel) for help with RNA editing analyses. The authors extend their gratitude to Prof. Musarò Dr. Gabriella Dorbowolny and Dr. Gaia Laurenzi (La Sapienza University), for providing SOD1$^{G93A}$ muscle lysates. Appreciation is also extended to Dr. Daniela Noain, Ines Antunes dos Santos Dias and Irena Barbaric (Department of Neurology, University of Zurich) for their kind contribution of DLB hindlimb skeletal muscles and Dr. Ruiqing Ni's group and Benjamin Francois Combes (Institute for Biomedical Engineering, University of Zurich) for providing DLB hindlimb skeletal muscles. We acknowledge the Tissu-Tumorotheque Est (CRB HCL, HCL's biobank) for providing the human biological samples (AD, DLB, ALS, FTD) used in this study.

## Author Contributions

**Conceptualization:** Silvia Sorce, Mario Nuvolone, Adriano Aguzzi.

**Data curation:** Davide Caredio, Maruša Koderman, Karl J. Frontzek.

**Formal analysis:** Davide Caredio, Maruša Koderman.

**Funding acquisition:** Karl J. Frontzek, Adriano Aguzzi.

**Investigation:** Davide Caredio, Maruša Koderman, Karl J. Frontzek.

**Methodology:** Davide Caredio, Maruša Koderman, Karl J. Frontzek, Silvia Sorce, Mario Nuvolone, Juliane Bremer, Petra Schwarz, Stefano Sellitto, Claudia Scheckel.

**Project administration:** Adriano Aguzzi.

**Resources:** Nathalie Streichenberger, Adriano Aguzzi.

**Supervision:** Karl J. Frontzek, Adriano Aguzzi.

**Validation:** Davide Caredio, Maruša Koderman, Giovanni Mariutti, Lidia Madrigal, Marija Mitrovic.

**Visualization:** Davide Caredio, Maruša Koderman.

**Writing – original draft:** Davide Caredio, Maruša Koderman, Karl J. Frontzek, Adriano Aguzzi.

**Writing – review & editing:** Davide Caredio, Maruša Koderman, Adriano Aguzzi.

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
