## [Editor Report · Decision Letter 0]

25 Jun 2024

Dear Prof. Aguzzi,

Thank you very much for submitting your manuscript "Prion diseases disrupt the glutamate/glutamine metabolism in skeletal muscle" for consideration at PLOS Pathogens. As with all papers reviewed by the journal, your manuscript was reviewed by members of the editorial board upon receipt and had also been independently reviewed by two reviewers via Review Commons. In light of the reviews you received and your indicated plans for revision, we would like to invite the resubmission of a significantly-revised version that takes into account each of the reviewers' comments in the manner you have described.

We cannot make any decision about publication until we have had the opportunity to review your revised manuscript and your response to the reviewers' comments. We also cannot exclude the possibility that your revised manuscript may be returned to the original set of reviewers for further evaluation.

Sincerely,

Neil A. Mabbott

Section Editor

PLOS Pathogens

Michael Malim

Editor-in-Chief

PLOS Pathogens

orcid.org/0000-0002-7699-2064
---

## [Editor Report · Decision Letter 1]

22 Aug 2024

Dear Prof. Aguzzi,

Thank you very much for submitting your manuscript "Prion diseases disrupt the glutamate/glutamine metabolism in skeletal muscle" for consideration at PLOS Pathogens. As with all papers reviewed by the journal, your manuscript was reviewed by members of the editorial board and by several independent reviewers. The reviewers appreciated the attention to an important topic. Based on the reviews, we are likely to accept this manuscript for publication, providing that you modify the manuscript according to the review recommendations.

Thank you for submitting your revised manuscript and for providing a detailed response to the suggestions made by the reviewers. Attention to the points raised has in my opinion improved the quality of the study. Before acceptance can I ask you to attend to the following minor text issues please?

-Title: is "the" necessary in the title?

-Materials & Methods: Page 23 (unfortunately line numbers not available) "Prions detection in the spleen" section. Can you check that the correct unit descriptors have been used in this section and throughout the M&Ms? Is it correct that 50 mg of spleen tissue was first homogenised in 500 mL (half a litre) of buffer? Also I noted use of "100 ug" instead of 100 micro in this section.

Sincerely,

Neil A. Mabbott

Section Editor

PLOS Pathogens

Michael Malim

Editor-in-Chief

PLOS Pathogens

orcid.org/0000-0002-7699-2064

Editor Comments:

Dear Prof. Aguzzi (Adriano),

Thank you for submitting your revised manuscript and for providing a detailed response to the suggestions made by the reviewers. Attention to the points raised has in my opinion improved the quality of the study. Before acceptance can I ask you to attend to the following minor text issues please?

-Title: is "the" necessary in the title?

-Materials & Methods: Page 23 (unfortunately line numbers not available) "Prions detection in the spleen" section. Can you check that the correct unit descriptors have been used in this section and throughout the M&Ms? Is it correct that 50 mg of spleen tissue was first homogenised in 500 mL (half a litre) of buffer? Also I noted use of "100 ug" instead of 100 micro in this section.

Figure Files:

Data Requirements:

Reproducibility:

References:

---

## [Editor Report · Decision Letter 2]

2 Sep 2024

Dear Prof. Aguzzi,

We are pleased to inform you that your manuscript 'Prion diseases disrupt glutamate/glutamine metabolism in skeletal muscle' has been provisionally accepted for publication in PLOS Pathogens.

Best regards,

Neil A. Mabbott

Section Editor

PLOS Pathogens

Neil Mabbott

Section Editor

PLOS Pathogens

Michael Malim

Editor-in-Chief

PLOS Pathogens

orcid.org/0000-0002-7699-2064
---

## [Editor Report · Acceptance letter]

4 Sep 2024

Dear Prof. Aguzzi,

We are delighted to inform you that your manuscript, "Prion diseases disrupt glutamate/glutamine metabolism in skeletal muscle," has been formally accepted for publication in PLOS Pathogens.

Best regards,

Michael Malim

Editor-in-Chief

PLOS Pathogens

orcid.org/0000-0002-7699-2064